# A Case for Automated Segmentation of MRI Data in Neurodegenerative Diseases: Type II GM1 Gangliosidosis

**DOI:** 10.3390/neurosci6020031

**Published:** 2025-04-03

**Authors:** Connor J. Lewis, Jean M. Johnston, Precilla D’Souza, Josephine Kolstad, Christopher Zoppo, Zeynep Vardar, Anna Luisa Kühn, Ahmet Peker, Zubir S. Rentiya, Muhammad H. Yousef, William A. Gahl, Mohammed Salman Shazeeb, Cynthia J. Tifft, Maria T. Acosta

**Affiliations:** 1Office of the Clinical Director and Medical Genetics Branch, National Human Genome Research Institute, 10 Center Drive, Bethesda, MD 20892, USA; connor.lewis@nih.gov (C.J.L.); jean.johnston@nih.gov (J.M.J.); cynthiat@mail.nih.gov (C.J.T.); 2McLean Hospital, Belmont, MA 02478, USA; jk068@bucknell.edu; 3Department of Radiology, University of Massachusetts Chan Medical School, Worcester, MA 01655, USA; czoppo@mgh.harvard.edu (C.Z.); zeynep.vardar@umassmed.edu (Z.V.); anna.kuhn@umassmemorial.org (A.L.K.); mohammed.shazeeb@umassmed.edu (M.S.S.); 4Koç University Hospital, Istanbul 34010, Türkiye; doktorpeker@gmail.com; 5Department of Radiation Oncology & Radiology, University of Virginia, Charlottesville, VA 22903, USA; zrentiya@rirrc.net; 6Department of Perioperative Medicine, National Institutes of Health Clinical Center, 10 Center Drive, Bethesda, MD 20892, USA; muhammad.yousef@nih.gov; 7Medical Genetics Branch, National Human Genome Research Institute, 10 Center Drive, Bethesda, MD 20892, USA; gahlw@mail.nih.gov

**Keywords:** GM1 gangliosidosis, imaging biomarkers, MRI segmentation

## Abstract

Background: Volumetric analysis and segmentation of magnetic resonance imaging (MRI) data is an important tool for evaluating neurological disease progression and neurodevelopment. Fully automated segmentation pipelines offer faster and more reproducible results. However, since these analysis pipelines were trained on or run based on atlases consisting of neurotypical controls, it is important to evaluate how accurate these methods are for neurodegenerative diseases. In this study, we compared five fully automated segmentation pipelines, including FSL, Freesurfer, volBrain, SPM12, and SimNIBS, with a manual segmentation process in GM1 gangliosidosis patients and neurotypical controls. Methods: We analyzed 45 MRI scans from 16 juvenile GM1 gangliosidosis patients, 11 MRI scans from 8 late-infantile GM1 gangliosidosis patients, and 19 MRI scans from 11 neurotypical controls. We compared the results for seven brain structures, including volumes of the total brain, bilateral thalamus, ventricles, bilateral caudate nucleus, bilateral lentiform nucleus, corpus callosum, and cerebellum. Results: We found volBrain’s vol2Brain pipeline to have the strongest correlations with the manual segmentation process for the whole brain, ventricles, and thalamus. We also found Freesurfer’s recon-all pipeline to have the strongest correlations with the manual segmentation process for the caudate nucleus. For the cerebellum, we found a combination of volBrain’s vol2Brain and SimNIBS’ headreco to have the strongest correlations, depending on the cohort. For the lentiform nucleus, we found a combination of recon-all and FSL’s FIRST to give the strongest correlations, depending on the cohort. Lastly, we found segmentation of the corpus callosum to be highly variable. Conclusions: Previous studies have considered automated segmentation techniques to be unreliable, particularly in neurodegenerative diseases. However, in our study, we produced results comparable to those obtained with a manual segmentation process. While manual segmentation processes conducted by neuroradiologists remain the gold standard, we present evidence to the capabilities and advantages of using an automated process that includes the ability to segment white matter throughout the brain or analyze large datasets, which pose feasibility issues to fully manual processes. Future investigations should consider the use of artificial intelligence-based segmentation pipelines to determine their accuracy in GM1 gangliosidosis, lysosomal storage disorders, and other neurodegenerative diseases.

## 1. Introduction

Volumetric analysis of magnetic resonance imaging (MRI) has proven useful in diagnosing and monitoring the progression of various neurological disorders [1,2]. The capability to evaluate brain growth and development has provided important insights in brain development and pathogenesis [3]. In Alzheimer’s disease, volumetric MRI analysis has been used to predict the progression of Alzheimer’s from mild cognitive impairment [4]. In Parkinson’s disease, volumetric MRI analysis has shown reduced putamen volume in Parkinson’s patients compared to controls [5]. In rarer diseases, volumetric MRI analysis has shown increased volumes of the total brain, white matter, and corpus callosum in neurofibromatosis type 1 [6]. In neuronal ceroid lipofuscinosis 3, volumetric MRI analysis has been used to demonstrate disease progression in the form of decreased supratentorial cortical gray matter and supratentorial white matter and cerebellar gray matter, as well as decreased volume in the basal ganglia, thalamus, and hippocampus [7]. Numerous techniques for performing these volumetric calculations exist, including both manual and automated methods [8,9].

Manual segmentation processing of MRI data involves a clinician manually tracing the structure of interest slice by slice. While these approaches are accurate, they are also time-consuming and unrealistic for analyzing larger datasets [10]. Inter-rater reliability issues are also noteworthy for manual approaches when results are compared between clinicians [11,12]. Automated segmentation processes have received increased attention with the capability of analyzing large datasets systematically without fatigue [13]. Automated segmentation of MRI data typically falls into one of two categories: either atlas-based or artificial intelligence (AI)-based methods [14]. Atlas-based techniques utilize either a hand-labeled or statistical atlas, which is registered to the scan being analyzed to identify neuroanatomical structures [15]. AI-based segmentation processes are trained to analyze neuroanatomical structures based on labeled training data and are then deployed to analyze MRI scans [16]. AI techniques are more flexible based on their training data and, once trained, may reduce the computational cost [17]. However, AI approaches have not had widespread adoption at the time of this study, and current approaches are more specialized [18,19,20,21,22].

Similarly, atlas-based segmentation processes have their own challenges. Firstly, the atlases utilized in the most common MRI segmentation pipelines may not be representative of the population being analyzed. For instance, Freesurfer utilizes the Desikan–Killiany dataset with 40 neurotypical control participants between the ages of 19 and 86 years, which may limit interpretation of subjects who are outside of this age range or who might be impaired [23]. Similarly, with the pediatric brain specifically, challenges arise, including increased noise, reduced contrast between tissues, and ongoing myelination [24]. Furthermore, some of the most utilized automated segmentation methods ignore smaller substructures, like the hypothalamus [25], pons [26], pituitary gland [27], and optic nerve [28]. Ultimately, it is important to investigate which atlas-based technique is suitable for each analysis. In this study, we aim to investigate some of the most frequently utilized atlas-based techniques, including Freesurfer, FSL, volBrain, and SPM, in GM1 gangliosidosis brains to understand which pipeline provides the highest accuracy of volumetric measurements compared to a manual approach.

GM1 gangliosidosis is an inherited ultra-rare neurodegenerative lysosomal storage disorder caused by variants in the GLB1 gene encoding β-galactosidase [29]. Three clinical subtypes for GM1 exist, based on the age of symptom onset and residual enzyme activity [30]. Type I (infantile) GM1 is the most severe form of the disease, with symptoms beginning in the first year of life, and death often occurring before the age of three [31]. Type II GM1 gangliosidosis can be further classified into two forms (late-infantile and juvenile), with the late-infantile form leading to symptom onset between age one and two, and survival into the second decade. Juvenile GM1 patients have symptom onset between three and five years old, with survival into the fourth decade [32]. Type III (adult) GM1 gangliosidosis patients typically have symptoms manifesting in the second decade, slower disease progression, and the longest survival. Currently, there are no approved therapies for GM1, and the disease is uniformly fatal. However, investigations into adeno-associated virus (AAV)-mediated gene therapies have shown promise in animal models, and a clinical trial with Type I and II GM1 gangliosidosis patients is currently underway (ClinicalTrials.gov identifier NCT03952637) [33,34,35].

Previous investigations utilizing brain MRI in Type II GM1 gangliosidosis patients have demonstrated numerous findings [36,37]. Atrophy of the cerebral cortex, corpus callosum, caudate nucleus, cerebellum, cerebellar white matter, basal ganglia, and associated ventricular enlargement have all been described [36,37]. Furthermore, in our previous study, we showed volumetric reductions in the volumes of the total brain, caudate nucleus, thalamic nucleus, lentiform nucleus, and corpus callosum, and ventricle enlargement that correlated with patients’ clinical severity and ability to meet the demands of daily life, emphasizing volume-based morphology’s role as a sensitive imaging marker in this cohort [38]. In this study, we first investigated the performance of five different fully automated MRI segmentation pipelines compared to a manual process in identifying MRI pathogenesis in GM1 gangliosidosis patients (Figure 1). Second, we evaluated which automated pipeline gave the most accurate results in terms of volumetric analysis of the different brain structures.

## 2. Methods

### 2.1. Type 2 GM1 Gangliosidosis Participants

Twenty-four GM1 patients from the “Natural History of Glycosphingolipid Storage Disorders and Glycoprotein Disorders” study (ClinicalTrials.gov ID: NCT00029965) were included in this study [39]. As described in D’Souza et al., a GM1 diagnosis was made through β-galactosidase enzyme deficiency or by biallelic variants in GLB1 [29]. Forty-five MRI scans from 16 juvenile (baseline age: 11.8 ± 4.9 years) patients were included in this study alongside 11 MRI scans from 8 late-infantile patients (Figure 2, baseline scan age: 5.5 ± 1.8 years, Appendix A).

### 2.2. Neurotypical Controls

Neurotypical early childhood control MRI scans were gathered from the Open Science Framework and included participants from the “Calgary preschool magnetic resonance imaging (MRI) dataset” and consisted of participants between 2 and 8 years of age recruited by the University of Calgary [40]. Late childhood and adolescent neurotypical control MRI data were gathered from Figshare and included participants from the “Detailing neuroanatomical development in late childhood and early adolescence using NODDI” study and consisted of participants between 8 and 13 years of age also recruited by the University of Calgary [41]. In total, 19 MRI scans were obtained from 11 neurotypical controls (Figure 2, baseline age: 8.3 ± 4.2 years).

### 2.3. T1-Weighted MRI Acquisition

MRI scans were performed on a Philips 3T system (Achieva, Philips Healthcare, Best, The Netherlands) for all GM1 patients under sedation at all time points with an 8-channel SENSE head coil. Images were acquired using a 3D T1-weighted protocol with a slice thickness of 1 mm, repetition time (TR) of 11 ms, and an echo time (TE) of 7 ms. Unprocessed digital imaging and communications in medicine (DICOM) images were converted to NIfTI using dcm2niix [42]. Calgary preschool and adolescent neurotypical controls were scanned without sedation on a General Electric 3T system (MR750w, GE Healthcare, Chicago, IL, USA) using a 32-channel head coil [40,41]. The T1-weighted images from the neurotypical preschool controls were acquired with TR/TE = 8.23/3.76 ms and a resolution of 0.9 mm × 0.9 mm × 0.9 mm (resampled to 0.45 mm × 0.45 mm × 0.9 mm) [34]. The T1-weighted images from the Calgary neurotypical adolescent controls were acquired with TR/TE = 8.21/3.16 ms and a 0.8 mm^3^ isotropic resolution [41].

### 2.4. Manual MRI Volumetric Segmentation

Volumetric segmentations were performed on the 3D T1-weighted images, which provided sufficient tissue contrast to visualize the structures of interest. A semi-automated approach was used, with manual corrections for the larger structures, and a manual process was implemented for the smaller structures, as previously described Zoppo et al. [12]. The following structures were segmented on all the scans: whole brain (without ventricles), cerebellum, ventricles, corpus callosum, caudate, lentiform nucleus, and thalamus. In brief, a team of researchers and trained neuroradiologists used the AMIRA analysis software (Amira, Thermo Fischer Scientific, Waltham, MA, USA) in the native image space to define the regions of interest (ROIs) around the boundaries of the structures using a combination of signal thresholding and/or manual demarcation on a slice-by-slice basis. The method for determining the segmentation boundaries of each structure was carried out as described in prior work [12]. The ROIs from all slices were rendered into a 3D volume for each structure to estimate the structure volume.

### 2.5. Freesurfer Volumetric Segmentation

Automated segmentation of MRI images was first performed using Freesurfer’s (v7.4.1) recon-all reconstruction pipeline to calculate volumes for the structures of interest [43,44,45,46,47,48,49,50,51]. The volumes of the whole brain (the sum of gray matter and white matter), ventricles, cerebellum, caudate, thalamus, lentiform nucleus (sum of the globus pallidus and putamen), and corpus callosum were calculated, corresponding to the manual segmentation process. Automated segmentation was performed post-hoc in the manual segmentation process.

### 2.6. FSL Volumetric Segmentation

MRI data was also segmented using tools from the FMRIB Software Library (FSL) [52,53,54]. The T1-weighted images were first sent through FSL’s BET for brain image extraction. Brain-extracted images were then analyzed using FSL’s FAST to create partial volume maps of the cerebrospinal fluid, gray matter, and white matter. T1-weighted images were also sent through FSL’s run_first_all pipeline to segment subcortical structures, including the bilateral thalamus, bilateral putamen, bilateral globus pallidus, and bilateral caudate nucleus. FSL’s fslstats was then used to calculate volumes from the partial volume maps and segmented images [52,53,54].

### 2.7. volBrain Volumetric Segmentation

Automated segmentation of MRI images was also performed using volBrain’s updated segmentation algorithm, vol2Brain [55]. vol2Brain uses the preprocessing algorithm from the original volBrain pipeline, followed by segmentation [56]. T1-weighted images in this study were uploaded to the volBrain server, and volumetric results were obtained for the intracranial volume, bilateral gray matter, bilateral white matter, cerebellum, bilateral thalamus, bilateral globus pallidus, bilateral putamen, and bilateral caudate nucleus.

### 2.8. SPM Volumetric Segmentation

Automated segmentation of T1-weighted images was also performed using statistical parametric mapping (SPM) 12, housed within MATLAB R2023a (The MathWorks Inc., Natick, MA, USA) [57]. Tissue probability maps of the cerebrospinal fluid, gray matter, and white matter were created using the SEGMENT tool. FSL’s fslstats was then used to calculate volumes from the tissue probability maps.

### 2.9. Headreco Volumetric Segmentation

The T1-weighted images in this study were also sent through SimNIBS’ (v3.2.6) headreco segmentation pipeline [58]. Headreco utilizes SPM12, followed by the computational anatomy toolbox (CAT12) to improve segmentation by SPM12 [57,58,59,60]. Gray matter, white matter, and cerebrospinal fluid volumes were calculated from the final mask generated by headreco using fslstats. Ventricle volume was also calculated from the ventricle mask generated by headreco using fslstats. Masks for the bilateral thalamus, corpus callosum, and cerebellum were generated from CAT12 after performing deformation-based morphometry (DBM), registering 555 healthy participants to the IXI-database [61], and using the Hammers atlas [62] to select ROIs. Volumes of the bilateral thalamus, corpus callosum, and cerebellum were calculated from each respective mask using fslstats.

### 2.10. Statistical Analysis

Cross-sectional data were analyzed in GraphPad Prism (v10.1.0, GraphPad Software, Boston, MA, USA) using Welch’s 2 sample *t*-test to demonstrate differences between neurotypical controls and both juvenile and late-infantile GM1 patients. Neurotypical controls were selected to age-match the late-infantile (Appendix A) and juvenile (Appendix A) GM1 patients separately since the juvenile patients were older than the late-infantile patients. MRI scans that failed to process are outlined in Appendix A. Linear regression modeling was performed in GraphPad Prism to calculate correlation coefficients (R^2^) and F-statistics; *p*-values from the F-statistics were compared between each automated segmentation process and the manual segmentation process for the 7 structures of interest.

## 3. Results

### 3.1. Whole Brain Volume

Manual whole brain volume segmentation demonstrated higher whole brain volumes in neurotypical controls when compared to both late-infantile (*t*(12) = 3.47, *p* = 0.0047, Figure 3) and juvenile (*t*(19) = 2.87, *p* = 0.0101, Figure 4) GM1 patients. volBrain, SPM, and headreco also demonstrated higher whole brain volumes in neurotypical controls compared to late-infantile patients; however, Freesurfer and FSL did not find a significant difference between late-infantile patients and neurotypical controls. Freesurfer, volBrain, and SPM demonstrated higher whole brain volumes in neurotypical controls compared to juvenile patients; however, FSL and headreco did not find a significant difference between the two cohorts (Figure 4).

Correlations between the five fully automated segmentation processes and our manual process showed volBrain (R^2^ = 0.9852) and FSL (R^2^ = 0.9058) R^2^ values above 0.90 in neurotypical controls (Figure 5). In juvenile GM1 patients, volBrain (R^2^ = 0.9583), Freesurfer (R^2^ = 0.9196), and SPM (R^2^ = 0.9642) all had R^2^ values above 0.90 (Figure 6). In late-infantile GM1 patients, volBrain (R^2^ = 0.9169) was the only fully automated segmentation process with an R^2^ above 0.90 when compared to the manual process (Figure 7).

### 3.2. Ventricle Volume

Manual ventricle volumetric segmentation demonstrated higher ventricle volume in late-infantile GM1 patients when compared to neurotypical controls (*t*(12) = 2.98, *p* = 0.0115, Figure 8). Freesurfer, volBrain, and headreco were also able to demonstrate this difference finding enlargement of the ventricles in late-infantile GM1 patients (Figure 8). In juvenile GM1 patients, no statistical difference was found in ventricle volume when compared to neurotypical controls using manual segmentation (*t*(19) = 2.98, *p* = 0.0115, Appendix A. Freesurfer, volBrain, and headreco also did not find a statistical difference in ventricle volume between juvenile GM1 patients and neurotypical controls (Appendix A).

Correlations between the three fully automated ventricle segmentation processes and our manual process showed volBrain (R^2^ = 0.9452) as the only pipeline with an R^2^ value above 0.90 in neurotypical controls (Figure 5). In juvenile GM1 patients, volBrain (R^2^ = 0.9782) and Freesurfer (R^2^ = 0.9765) both had R^2^ values above 0.90 (Figure 6). In late-infantile GM1 patients, volBrain (R^2^ = 0.9974) and headreco (R^2^ = 0.9950) both had R^2^ values above 0.90 when compared to the manual process (Figure 7).

### 3.3. Cerebellar Volume

Manual cerebellar volume segmentation demonstrated higher cerebellar volumes in neurotypical controls when compared to both late-infantile (*t*(12) = 3.32, *p* = 0.0061, Appendix A) and juvenile (*t*(19) = 2.69, *p* = 0.0146, Appendix A) GM1 patients. In late-infantile GM1 patients, Freesurfer, volBrain, and headreco also demonstrated this result, finding larger cerebellar volumes in neurotypical controls (Appendix A). In juvenile GM1 patients, Freesurfer and volBrain also demonstrated this result, finding larger cerebellar volumes in neurotypical controls; however, headreco did not find a statistical difference between juvenile GM1 patients and neurotypical controls (Appendix A).

Correlations between the three fully automated cerebellar segmentation processes and our manual process showed that volBrain (R^2^ = 0.8259) was the only pipeline with an R^2^ value above 0.80 in neurotypical controls (Figure 5). In juvenile GM1 patients, volBrain (R^2^ = 0.9258) and Freesurfer (R^2^ = 0.9133) both had R^2^ values above 0.90 (Figure 6). In late-infantile GM1 patients, volBrain (R^2^ = 0.8374) and headreco (R^2^ = 0.8779) both had R^2^ values above 0.80 when compared to the manual process (Figure 7).

### 3.4. Thalamic Volume

Manual thalamic volume segmentation demonstrated higher thalamic volumes in neurotypical controls when compared to both late-infantile (*t*(12) = 6.14, *p* < 0.0001, Appendix A) and juvenile (*t*(19) = 3.51, *p* = 0.0023, Figure 9) GM1 patients. In late-infantile GM1 patients, Freesurfer, FSL, volBrain, and headreco also demonstrated this result, finding larger thalamic volumes in neurotypical controls (Appendix A). In juvenile GM1 patients, Freesurfer, volBrain, and headreco also demonstrated this result, finding larger thalamic volumes in neurotypical controls; however, FSL did not find a statistical difference between juvenile GM1 patients and neurotypical controls (Figure 9).

Correlations between the four fully automated thalamic segmentation processes and our manual process showed that volBrain (R^2^ = 0.6276) had the highest R^2^ value in neurotypical controls (Figure 5). In juvenile GM1 patients, volBrain (R^2^ = 0.8053) also had the highest R^2^ value (Figure 6). In late-infantile GM1 patients, volBrain (R^2^ = 0.8632) again had the highest R^2^ value when compared to the manual process (Figure 7).

### 3.5. Caudate Nucleus Volume

Manual caudate volume segmentation demonstrated higher thalamic volumes in neurotypical controls when compared to both late-infantile (*t*(12) = 2.72, *p* = 0.0185, Appendix A) and juvenile (*t*(19) = 6.00, *p* < 0.0001, Appendix A) GM1 patients. In late-infantile GM1 patients, Freesurfer, FSL, and volBrain also demonstrated this result, finding larger caudate volumes in neurotypical controls (Appendix A). In juvenile GM1 patients, Freesurfer also demonstrated this result, finding larger caudate volumes in neurotypical controls; however, FSL and volBrain did not find a statistical difference between juvenile GM1 patients and neurotypical controls (Appendix A).

Correlations between the three fully automated caudate nucleus segmentation processes and our manual process showed that Freesurfer (R^2^ = 0.8174) had the highest R^2^ value in neurotypical controls (Figure 5). In juvenile GM1 patients, volBrain (R^2^ = 0.8885) had the highest R^2^ value (Figure 6). In late-infantile GM1 patients, Freesurfer (R^2^ = 0.8536) had the highest R^2^ value when compared to the manual process (Figure 7).

### 3.6. Lentiform Nucleus Volume

Manual caudate volume segmentation demonstrated higher thalamic volumes in neurotypical controls when compared to both late-infantile (*t*(12) = 4.46, *p* = 0.0008, Appendix A) and juvenile (*t*(19) = 7.51, *p* < 0.0001, Appendix A) GM1 patients. In late-infantile GM1 patients, Freesurfer, FSL, and volBrain also demonstrated this result, finding larger lentiform nucleus volumes in neurotypical controls (Appendix A). In juvenile GM1 patients, Freesurfer, FSL, and volBrain also demonstrated this result, finding larger lentiform nucleus volumes in neurotypical controls (Appendix A).

Correlations between the three fully automated lentiform nucleus segmentation processes and our manual process showed that Freesurfer (R^2^ = 0.7251) had the highest R^2^ value in neurotypical controls (Figure 5). In juvenile GM1 patients, volBrain (R^2^ = 0.6605) had the highest R^2^ value (Figure 6). In late-infantile GM1 patients (Figure 7), Freesurfer (R^2^ = 0.6619) again had the highest R^2^ value when compared to the manual process. In late-infantile GM1 patients, volBrain had a similar correlation coefficient strength to Freesurfer (0.6518).

### 3.7. Corpus Callosum Volume

Manual corpus callosum volume segmentation demonstrated higher corpus callosum volumes in neurotypical controls when compared to both late-infantile (*t*(12) = 4.46, *p* = 0.0008, Appendix A) and juvenile (*t*(19) = 2.49, *p* = 0.0222, Appendix A) GM1 patients. In late-infantile GM1 patients, Freesurfer also demonstrated this result, finding larger corpus callosum volumes in neurotypical controls (Appendix A); however, headreco did not demonstrate this difference. In juvenile GM1 patients, headreco also demonstrated this result, finding larger corpus callosum volumes in neurotypical controls (Appendix A); however, Freesurfer did not demonstrate this difference.

Correlations between the two fully automated corpus callosum segmentation processes and our manual process showed that SimNIBS’ headreco (R^2^ = 0.4829) had the highest R^2^ value in neurotypical controls (Figure 5). In juvenile GM1 patients, Freesurfer (R^2^ = 0.4050) again had the highest R^2^ value (Figure 6). In late-infantile GM1 patients, both Freesurfer and headreco had weak correlations (R^2^ < 0.1) with the manual process (Figure 7).

### 3.8. Extended Results

Individual figures for the comparisons between juvenile and late-infantile neurotypical controls for the remaining analyzed structures are included in Appendix A, respectively. Tables with *p*-values for the correlation analysis between the manual and automated segmentation process (Figure 5, Figure 6 and Figure 7) are included in Appendix A. Individual figures for the correlations between each of the five automated pipelines with the manual process for all analyzed structures are included in Appendix A. Slope estimates and y-intercept values with standard errors for the correlations between each of the five automated pipelines with the manual process are included in Appendix A.

## 4. Discussion

This study assessed the capabilities of five fully automated brain MRI segmentation pipelines compared to a manual approach in juvenile and infantile GM1 gangliosidosis patients and neurotypical controls. We aimed to demonstrate the accuracy of the fully automated techniques, which have significant time-saving advantages and can enable the analysis of large-scale datasets and larger complex brain structures, including the cerebral white matter. We found our automated results closely reflected those found by the manual approach, suggesting the utility of fully automated processes in neurodegenerative diseases.

We first compared each automated segmentation pipeline’s ability to differentiate between GM1 patients and neurotypical controls using the seven brain structures of interest. The manual segmentation process found statistically significant differences between juvenile GM1 patients and neurotypical controls in six out of the seven brain structures. For the specific segmentations that were performed by each of the fully automated pipelines, statistically significant differences in brain structure volume were observed in five out of six for Freesurfer, four out of five for volBrain, one out of four for FSL, one out of one for SPM12, and two out of four for headreco. In late-infantile patients, the manual segmentation process found statistically significant differences in all seven brain structures. Freesurfer, volBrain, FSL, SPM12, and headreco identified statistically significant differences in six out of seven, six out of six, three out of four, one out of one, and four out of five brain structure volumes, respectively.

Correlation measurements were then performed, comparing the volumetric results from each of the fully automated pipelines with the results from the manual approach to understand how they relate to a clinician’s measurements. volBrain showed the strongest correlation with the manual pipeline for the whole brain, ventricles, and thalamus for all three cohorts (Figure 5, Figure 6 and Figure 7). volBrain also showed the strongest correlation in the cerebellum in both the neurotypical controls and juvenile GM1 patients, with headreco demonstrating the strongest correlation in the late-infantile GM1 patients. Freesurfer showed the strongest correlation with the manual pipeline for the caudate nucleus in neurotypical controls and late-infantile GM1 patients, with volBrain demonstrating the strongest correlation in the juvenile GM1 patients. Freesurfer also showed the strongest correlation for the lentiform nucleus in the neurotypical controls, with FSL demonstrating the strongest correlation in the late-infantile GM1 patients, and volBrain demonstrating the strongest correlation in the juvenile GM1 patients. Overall, our results suggest that among the five automated pipelines, volBrain is the most accurate for analyzing ventricle size, total brain volume, and thalamic volume in GM1 patients, while Freesurfer is the most accurate for analyzing the caudate nucleus.

Inter-structure variability in the accuracy of these measurements was observed. Segmentation of deep brain structures (thalamus, caudate nucleus, and lentiform nucleus) demonstrated reduced accuracy compared to the total brain, lateral ventricles, and cerebellum. The most likely explanation for this result centers around the relative signal intensity contrast between the analyzed structures and surrounding tissues. More specifically, the lateral ventricles, which are filled with cerebrospinal fluid, appear dark on a T1-weighted image compared to the surrounding tissue, offering a strong contrast [63]. Similarly, for measurements of total brain volume and cerebellar volume, the strong contrast is also present since both structures are surrounded by cerebrospinal fluid [64]. However, the complex structures of the gyri and folia likely reduce the accuracy of measurements for the brain and cerebellum compared to the lateral ventricles, respectively. Signal intensity contrast is reduced in deep brain structures. For instance, the thalamus, caudate nucleus, and both the globus pallidus and putamen (part of the lentiform nucleus) are primarily gray matter structures, which are more difficult to segment from the surrounding white matter regions [65,66,67,68]. This phenomenon is emphasized in the lateral borders of the thalamus and posterior borders of the globus pallidus, which have previously been described [69,70].

Corpus callosum segmentation in this study resulted in varying success. On the one hand, Freesurfer was able to demonstrate the correct positive relationship in the late-infantile GM1 patients, and headreco did so in juvenile GM1 patients. However, correlations with the manual process were weak and highly variable for all three cohorts. Furthermore, variability in the measured size was also observed. Headreco appeared to measure further into the bilateral hemispheres, while Freesurfer appeared to exclude relevant regions that were included in our manual process. This result is consistent with previous analyses that suggest that a dedicated corpus callosum segmentation process may be justified for future studies [71,72,73].

For more common neurodegenerative diseases like Alzheimer’s and Parkinson’s, fully automated segmentation of MRI data utilizing Freesurfer, CAT12, or SPM12 are commonplace [61,74,75,76]. However, this has not trickled down into rarer diseases, where segmentation techniques are also applicable for evaluating disease progression and potential therapeutic benefits. For instance, automated segmentation would be amenable to juvenile and late-infantile GM2 gangliosidosis, which has a similar MRI progression to juvenile and late-infantile GM1 gangliosidosis [37]. Furthermore, other lysosomal diseases, like Gaucher [77], Fabry [78], and neuronal ceroid lipofuscinoses [79,80], could be analyzed using this type of neuroimaging approach due to previously established brain involvement. This ultimately extends to cerebellar ataxias, Parkinsonism, and inherited prion diseases [81]. With this in mind, researchers and clinicians should evaluate the data of patients less than 2 years old with caution due to immature myelination [82]. Studies aiming to evaluate this infant patient population should consider Infant Freesurfer or alternative pipelines specifically designed for analyzing infant MRI data [83].

Limitations of this study need to be considered before the results are used to guide clinical practice.

First, this study is limited by a small sample size in neurotypical controls (*n* = 11, 19 MRI scans), juvenile GM1 patients (*n* = 16, 45 MRI scans), and late-infantile patients (*n* = 8, 11 MRI scans). However, this study represents the largest MRI dataset of Type II GM1 gangliosidosis patients [38]. Second, this study is limited by variations in the scanners and protocols used between the GM1 patients and neurotypical controls. However, all six analysis techniques were given the same images to analyze, and the results of this study focus on the comparisons between these techniques. Ultimately, future investigations should utilize identical protocols when comparing MRI segmentation techniques. Dice similarity coefficients, Hausdorff distance, and the mean distance to agreement were also not considered in this study [84,85,86]. Future studies investigating the accuracy of automated MRI segmentation methods should incorporate these metrics; however, in this pilot study, we focused on the capability of these methods for demonstrating GM1-associated neurodegeneration compared to neurotypical controls.

## 5. Conclusions

In this study, we compared five fully automated segmentation pipelines with a manual pipeline in neurotypical controls and juvenile and late-infantile GM1 gangliosidosis patients. We analyzed seven brain structures of interest, including the volumes of the total brain, cerebellum, ventricles, bilateral thalami, bilateral lentiform nucleus, bilateral caudate nucleus, and corpus callosum. We found that a combination of pipelines led to the strongest correlations with the manual pipeline, with volBrain’s vol2Brain demonstrating the strongest correlations for the whole brain, ventricles, and thalamus; Freesurfer’s recon-all demonstrating the strongest correlations for the caudate nucleus; and a combination of pipelines demonstrating the strongest correlations for the cerebellum and lentiform nucleus. Ultimately, we found our fully automated results to be consistent with our manual process; however, further studies are needed to investigate other neurodegenerative and lysosomal storage disorders.

## Figures and Tables

**Figure 1 neurosci-06-00031-f001:**
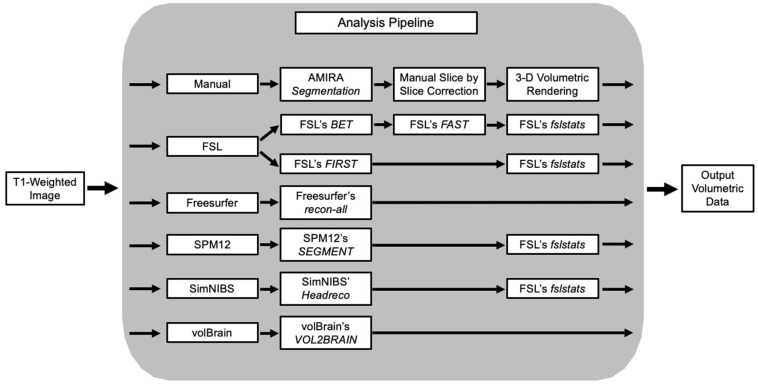
Segmentation analysis pipeline overview. T1-weighted MRI scans were processed through five fully automated segmentation pipelines in addition to the manual segmentation process. FSL required two different sub-pipelines. FSL’s FIRST was used to analyze bilateral volumes of the thalamus, caudate nucleus, and lentiform nucleus, and FSL’s FAST was used to analyze whole brain volumes.

**Figure 2 neurosci-06-00031-f002:**
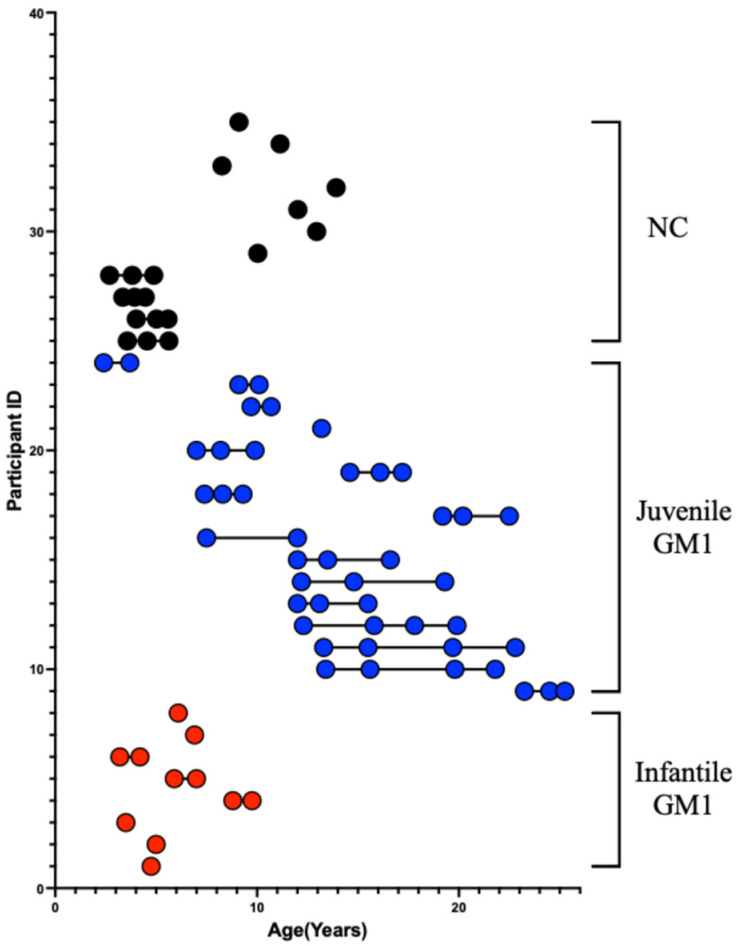
Age stratification of each participant at the T1-weighted MRI scan. Infantile GM1 patients are shown in red. Juvenile GM1 patients are shown in blue. Neurotypical controls are shown in black. Connecting lines indicate that the participant had repeated MRI scans.

**Figure 3 neurosci-06-00031-f003:**
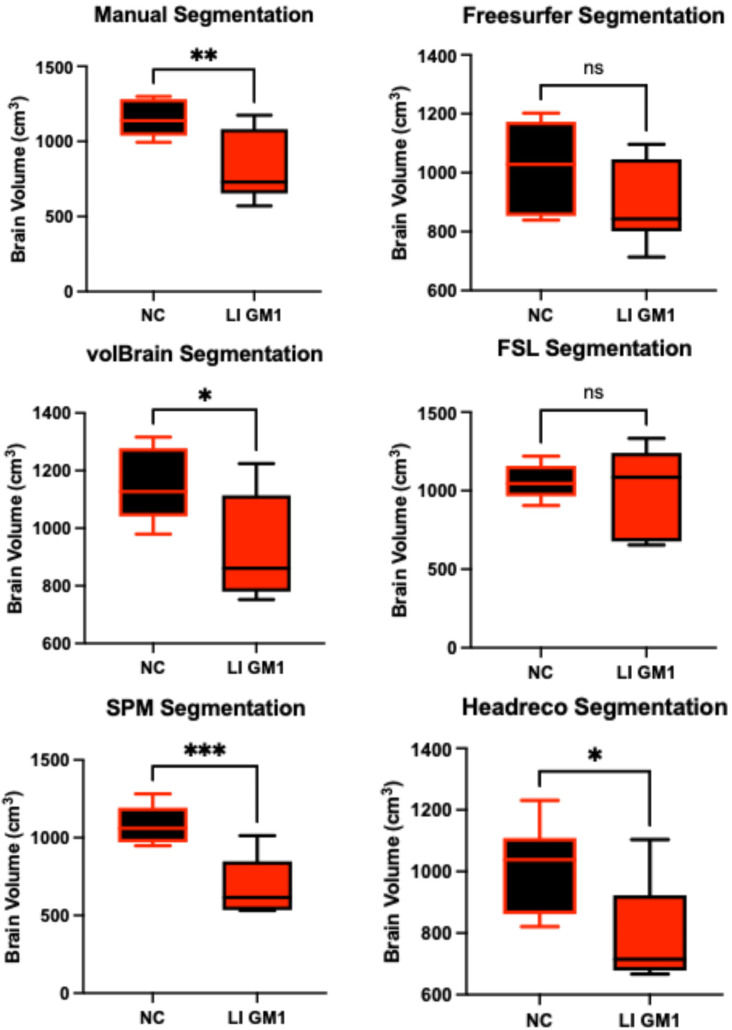
Cross-sectional evaluation of the five automated segmentation algorithms demonstrating cohort differences in total brain volume. Late-infantile (LI) GM1 patients (*n* = 7) are shown in red. Neurotypical controls (NC, *n* = 7) are shown in black. *p*-values were calculated from the *t*-statistic. * *p* < 0.05, ** *p* < 0.01, *** *p* < 0.001, and not significant (ns) for *p* > 0.05.

**Figure 4 neurosci-06-00031-f004:**
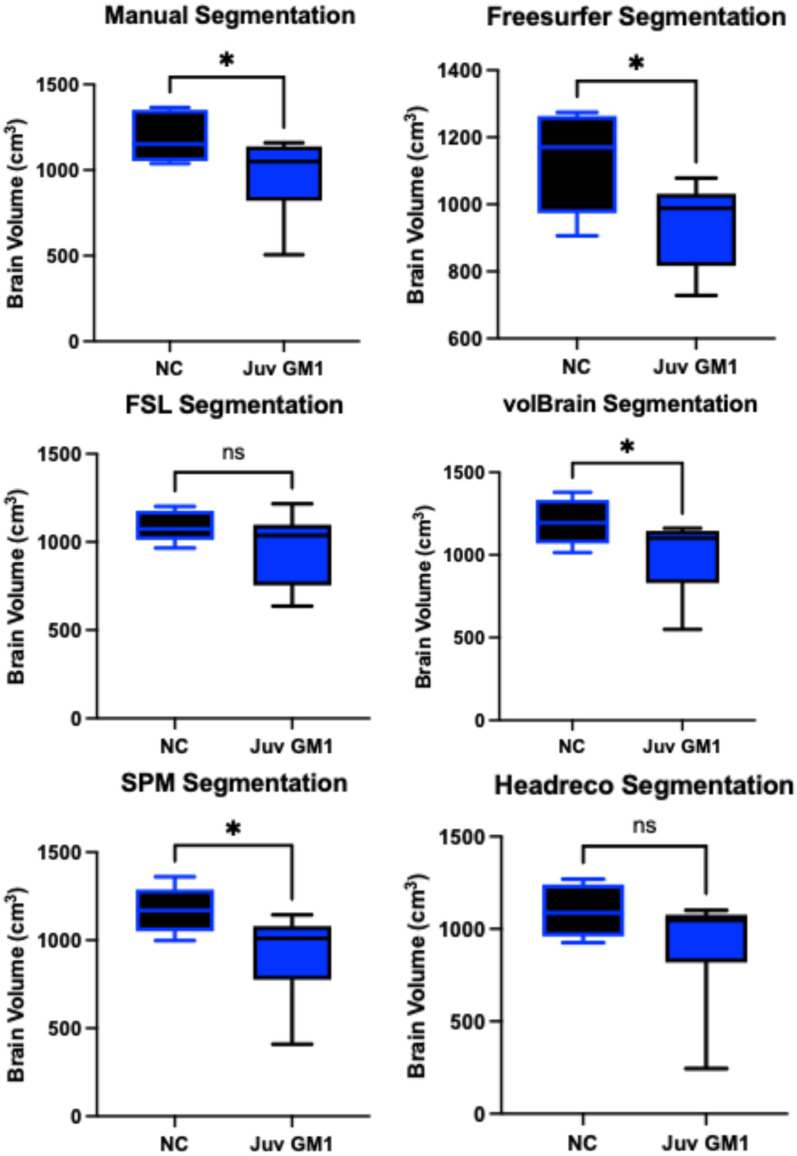
Cross-sectional evaluation of the five automated segmentation algorithms’ ability to demonstrate cohort differences in total brain volume. Juvenile (Juv) GM1 patients (*n* = 14) are shown in blue. Neurotypical controls (NC, *n* = 7) are shown in black. *p*-values were calculated from the *t*-statistic. * *p* < 0.05, not significant (ns) for *p* > 0.05.

**Figure 5 neurosci-06-00031-f005:**
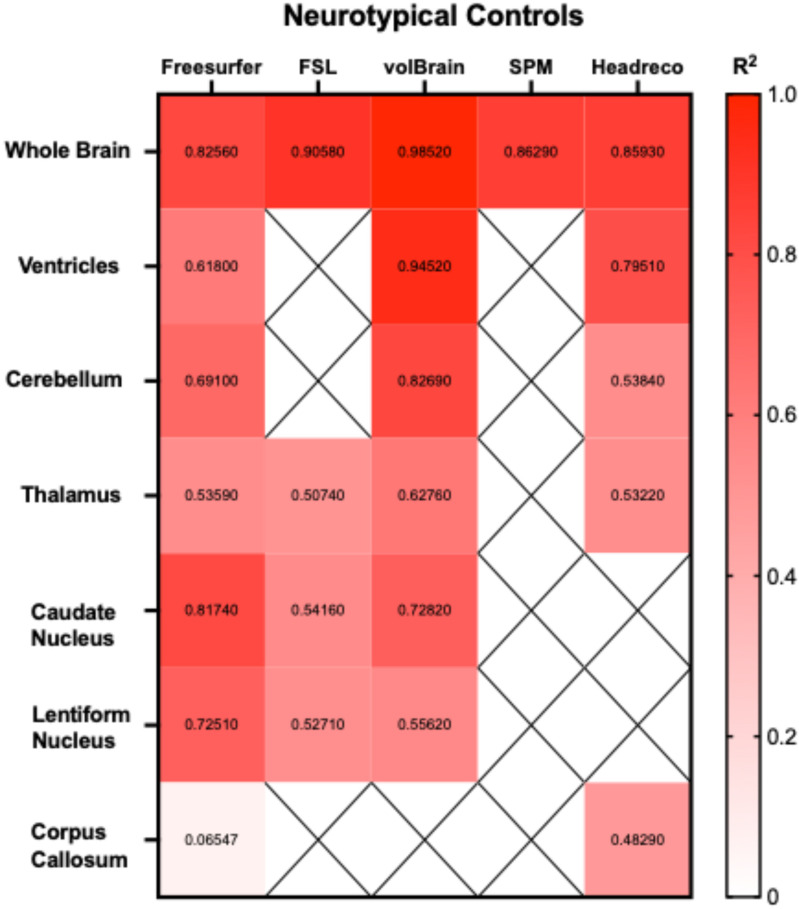
Heatmap of correlation strengths (R^2^) between the manual segmentation process and the five fully automated pipelines for the seven structures of interest in neurotypical controls. ‘X’ are designations where the region was not calculated using the specified segmentation algorithm.

**Figure 6 neurosci-06-00031-f006:**
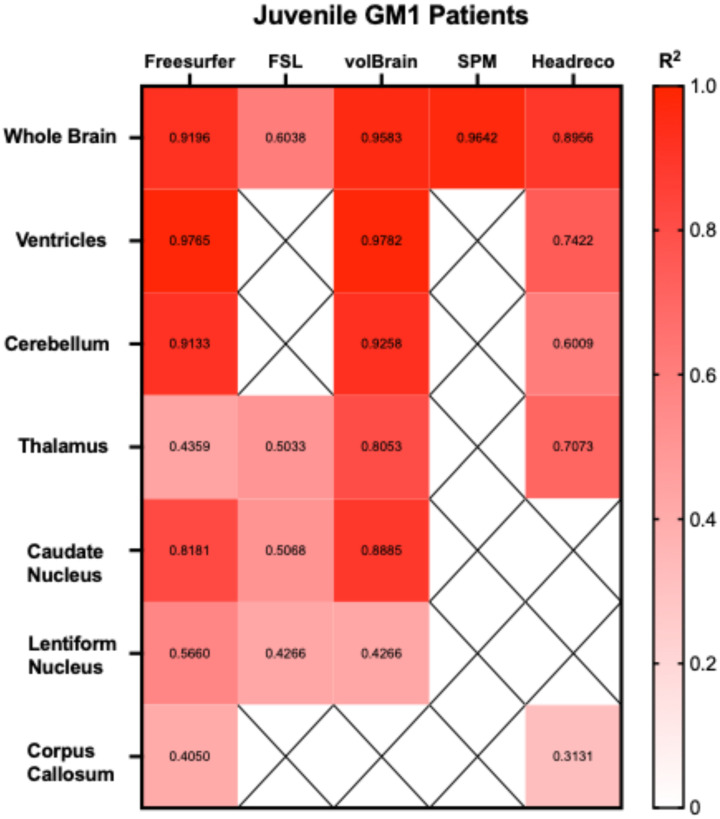
Heatmap of correlation strengths (R^2^) between the manual segmentation process and the five fully automated pipelines for the seven structures of interest in juvenile (Juv) GM1 patients. ‘X’ are designations where the region was not calculated using the specified segmentation algorithm.

**Figure 7 neurosci-06-00031-f007:**
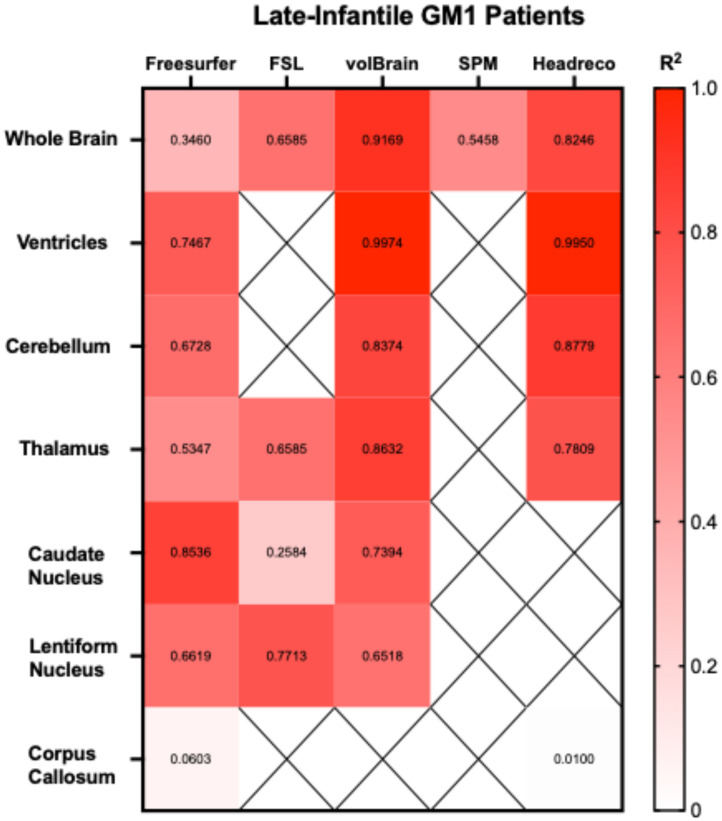
Heatmap of correlations strengths (R^2^) between the manual segmentation process and the five fully automated pipelines for the seven structures of interest in late-infantile (LI) GM1 patients. ‘X’ are designations where the region was not calculated using the specified segmentation algorithm.

**Figure 8 neurosci-06-00031-f008:**
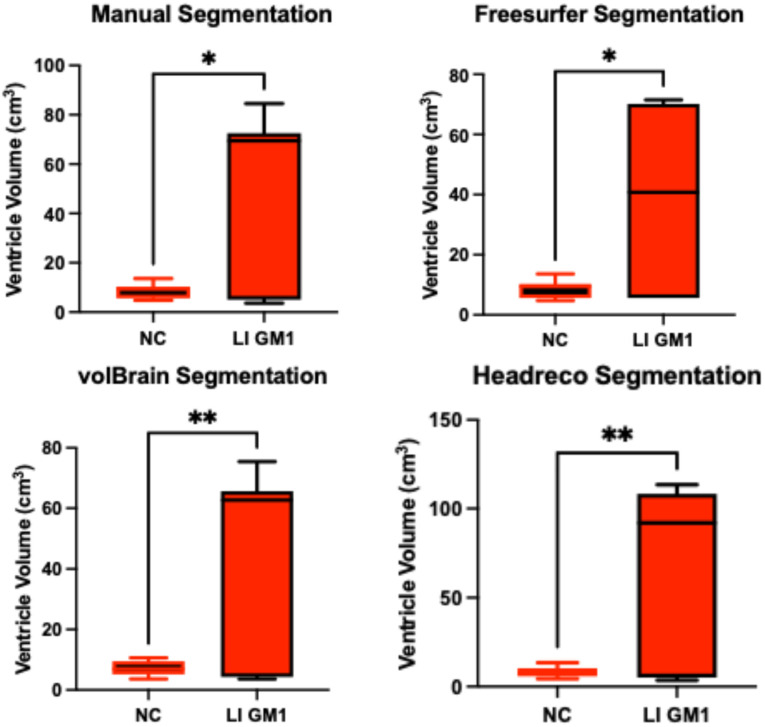
Cross-sectional evaluation of three automated segmentation algorithms demonstrating cohort differences in ventricle volume. Late-infantile (LI) GM1 patients (*n* = 7) are shown in red. Neurotypical controls (NC, *n* = 7) are shown in black. *p*-values were calculated from the *t*-statistic. * *p* < 0.05, ** *p* < 0.01.

**Figure 9 neurosci-06-00031-f009:**
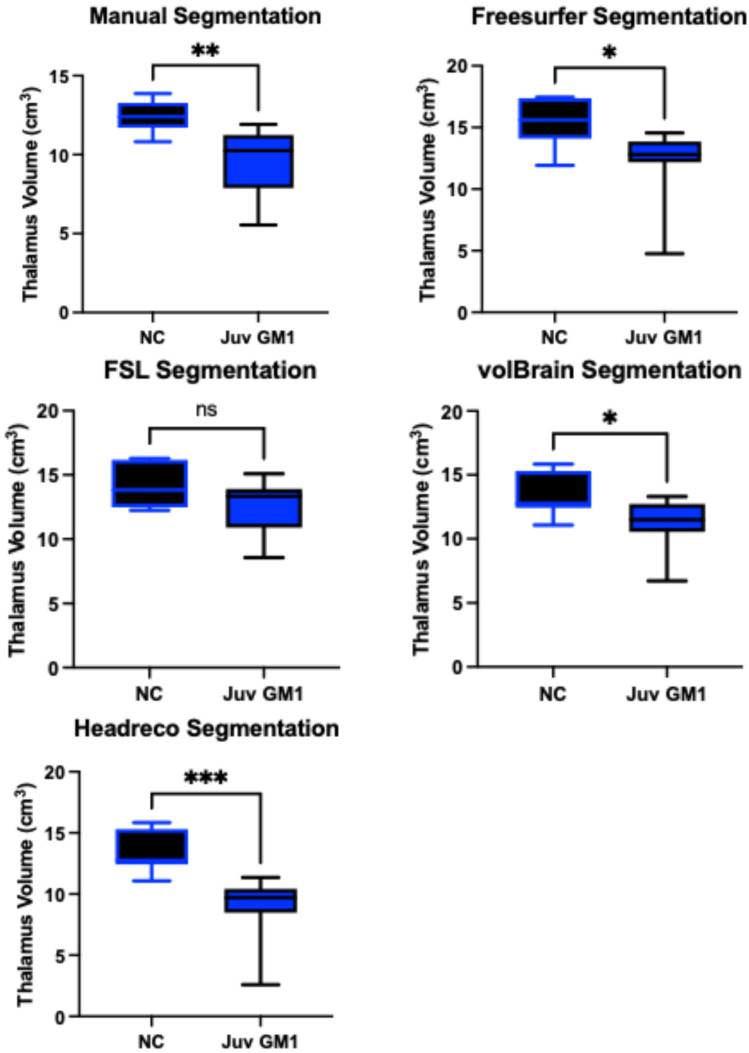
Cross-sectional evaluation of four automated segmentation algorithms’ ability to demonstrate cohort differences in thalamic volume. Juvenile (Juv) GM1 patients (*n* = 14) are shown in blue. Neurotypical controls (NC, *n* = 7) are shown in black. *p*-values were calculated from the *t*-statistic. * *p* < 0.05, ** *p* < 0.01, *** *p* < 0.001, and not significant (ns) for *p* > 0.05.

## Data Availability

The data described in this manuscript are available from the corresponding author upon reasonable request. Neuroimaging data for the early childhood neurotypical control group are publicly available here: https://osf.io/axz5r/ (accessed on 14 August 2024) [40]. Neuroimaging data for the adolescent neurotypical control group are publicly available here: https://doi.org/10.6084/m9.figshare.6002273.v2 (accessed on 15 August 2024) [41].

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
