# Peer review of "A Case for Automated Segmentation of MRI Data in Neurodegenerative Diseases: Type II GM1 Gangliosidosis"

_neurosci, 2025, doi:10.3390/neurosci6020031_

Round 1
Reviewer 1 Report
Comments and Suggestions for Authors
The title presents A Case for Automated Segmentation of MRI Data in Milder Neurodegenerative Diseases. However, GM1 gangliosidosis is a rare disease that falls into the category of severe neurodegenerative disorders. In particular, Type I (infantile) is highly fatal, and Type II (juvenile) is also a progressive disease that severely impairs neural function. Given this, I question whether it is appropriate to describe GM1 gangliosidosis as a milder neurodegenerative disease.
Additionally, while the study presents results specific to GM1 gangliosidosis, I recommend broadening the discussion to explore its applicability to neurodegenerative diseases in general.
The figures occupy an excessive number of pages. Adjusting the overall figure size and resolution is necessary.
In the introduction, I recommend adding a more detailed explanation of Automated Segmentation of MRI, including an overview of the current state of this technology. Additionally, the discussion section should incorporate a more in-depth comparison with existing segmentation methods.
The paper should include an introduction to the disease being studied. It would be beneficial to highlight the importance of MRI data and the significance of analytical methods in GM1 gangliosidosis, as well as their broader relevance to neurodegenerative diseases.
In the discussion, a more in-depth analysis should be included on why the automated MRI segmentation methods used in this study showed high correlations for specific brain structures.
Comments on the Quality of English LanguageNo specific comments.
Author Response
We would like to thank you for the acknowledgement of our work and appreciate the suggestions of how to improve our study.
Comment 1: The title presents A Case for Automated Segmentation of MRI Data in Milder Neurodegenerative Diseases. However, GM1 gangliosidosis is a rare disease that falls into the category of severe neurodegenerative disorders. In particular, Type I (infantile) is highly fatal, and Type II (juvenile) is also a progressive disease that severely impairs neural function. Given this, I question whether it is appropriate to describe GM1 gangliosidosis as a milder neurodegenerative disease.
Response 1: We appreciate Reviewer 1’s time and consideration of our manuscript. We agree that GM1 as a whole is a severely progressive neurodegenerative disease. The word ‘milder’ was designated in relation to Type I GM1 gangliosidosis patients, but this study focuses solely on Type II patients. However, we acknowledge this is a relative comparison and have removed the word milder in the Title of the manuscript. The title now reads “A Case for Automated Segmentation of MRI Data in Neurodegenerative Diseases: Type II GM1 Gangliosidosis”.
Comment 2: Additionally, while the study presents results specific to GM1 gangliosidosis, I recommend broadening the discussion to explore its applicability to neurodegenerative diseases in general.
We agree and have updated the discussion to include potential future use cases for this type of analysis with the following language: “For more common neurodegenerative diseases like Alzherimer’s and Parkinson’s, fully automated segmentation of MRI data utilizing Freesurfer, CAT12, or SPM12 are commonplace. However, this has not trickled down into rarer diseases where segmentation techniques are also applicable for evaluating disease progression and potential therapeutic benefit. For instance, automated segmentation would be amenable to juvenile and late-infantile GM2 gangliosidosis which has similar MRI progression to juvenile and late-infantile GM1 gangliosidosis. Furthermore, other lysosomal diseases like Gaucher, Fabry, and neuronal ceroid lipofuscinoses could be analyzed using this type of neuroimaging approach due to previously established brain involvement. This ultimately extends to cerebellar ataxias, Parkinsonism’s, and inherited prion diseases. With this in mind, researchers and clinicians should evaluate data of patients less than 2 years old with caution due to immature myelination. Studies aiming to evaluate this infant patient population should consider Infant Freesurfer or alternative pipelines specifically designed for analyzing infant MRI data.”
Comment 3: The figures occupy an excessive number of pages. Adjusting the overall figure size and resolution is necessary.
Response 3: We understand that the figures occupy a significant portion of the manuscript. However, we believe they are important visual tools for the understanding of our work. We are willing to move some figures to the supplementary materials if the editorial team deems it necessary.
Comment 4: In the introduction, I recommend adding a more detailed explanation of Automated Segmentation of MRI, including an overview of the current state of this technology. Additionally, the discussion section should incorporate a more in-depth comparison with existing segmentation methods.
Response 4: We have reviewed our manuscript and ultimately believe our paper provides a detailed explanation of the current state of automated MRI segmentation including capabilities, challenges, techniques, and tools. We believe this is reflected in the text from the manuscript below.
“Automated segmentation processes have received increased attention with the capability of analyzing large datasets systematically without fatigue. Automated segmentation of MRI data typically falls into one of two categories, either atlas-based or artificial intelligence (AI)-based methods. Atlas-based techniques utilize either a hand-labeled or statistical atlas, which is registered to the scan being analyzed to identify neuroanatomical structures. AI-based segmentation processes are trained to analyze neuroanatomical structures based on labeled training data and are then deployed to analyze MRI scans. AI techniques are more flexible based on their training data and once trained may reduce the computational cost. However, AI approaches have not had widespread adoption at the time of this study, and current approaches are more specialized.
Similarly, atlas-based segmentation processes have their own challenges. First, the atlases utilized in the most common MRI segmentation pipelines may not be representative of the population being analyzed. For instance, Freesurfer utilizes the Desikan-Killiany dataset with 40 neurotypical control participants between the ages of 19 and 86 years, which may limit interpretation of subjects who are outside of this age range or who might be impaired. Similarly, with the pediatric brain specifically, challenges arise including increased noise, reduced contrast between tissues, and ongoing myelination. Furthermore, some of the most utilized automated segmentation methods ignore smaller substructures like the hypothalamus, pons, pituitary gland, and optic nerve. Ultimately, it is important to investigate which atlas-based technique is suitable for each analysis. In this study, we aim to investigate some of the most frequently utilized atlas-based techniques including Freesurfer, FSL, volBrain, and SPM in GM1 gangliosidosis brains, to understand which pipeline provides the highest accuracy of volumetric measurements compared to a manual approach.”
Comment 5: The paper should include an introduction to the disease being studied. It would be beneficial to highlight the importance of MRI data and the significance of analytical methods in GM1 gangliosidosis, as well as their broader relevance to neurodegenerative diseases.
Response 5: We agree that the paper’s introduction should be broadened to include the significance of MRI volumetric analysis in neurodegenerative diseases. We have added the following language to address this concern. “In Alzheimer’s disease, volumetric MRI analysis has been used to predict the progression of Alzheimer’s from mild cognitive impairment. In Parkinson’s disease, volumetric MRI analysis has shown reduced Putamen volume in Parkinson’s patients compared to controls. In rarer diseases, volumetric MRI analysis has shown increased volumes of the total brain, white matter, and corpus callosum in neurofibromatosis type 1. In neuronal ceroid lipofuscinosis 3, volumetric MRI analysis has been used demonstrate disease progression in the form of decreased supratentorial cortical gray matter and supratentorial white matter, cerebellar gray matter, basal ganglia, thalamus, and hippocampus.”
We also agree that the significance of MRI data in GM1 gangliosidosis needs to be added to the introduction. To address this concern, we have added the following language: “Furthermore, in our previous study, we showed volumetric reductions in the volumes of the total brain, caudate nucleus, thalamic nucleus, lentiform nucleus, corpus callosum, and ventricle enlargement correlated with patients’ clinical severity and ability to meet the demands of daily life, emphasizing volume-based morphology’s role as a sensitive imaging marker in this cohort.”
Comment 6: In the discussion, a more in-depth analysis should be included on why the automated MRI segmentation methods used in this study showed high correlations for specific brain structures.
Response 6: We agree with the reviewer’s comments that a paragraph should be dedicated to discussing the accuracy of variations between structures. We have added the following language into the discussion to address this concern:
“Inter-structure variability in the accuracy of these measurements was observed. Segmentation of deep brain structures (thalamic, caudate, and lentiform nuclei) demonstrated reduced accuracy compared to the total brain, lateral ventricles, and cerebellum. The most likely explanation for this result centers around the relative signal intensity contrast between the analyzed structures and surrounding tissue. More specifically, the lateral ventricles which are filled with cerebrospinal fluid appear dark on a T1-weighted compared to surrounding tissue offering a strong contrast. Similarly, for measurements of total brain volume and cerebellar volume this is also present since both structures are surrounded by cerebrospinal fluid. However, the complex structures of the gyri and folia likely reduce the accuracy of measurements for the brain and cerebellum compared to the lateral ventricles, respectively. Signal intensity contrast is reduced in deep brain structures. For instance, the thalamus, caudate nucleus, and both the globus pallidus and putamen (part of the lentiform nucleus) are primarily gray matter structures which are more difficult to segment from the surrounding white matter regions. This phenomenon is emphasized in the lateral borders of the thalamus, and posterior borders of the globus pallidus, which have previously been described.”
Reviewer 2 Report
Comments and Suggestions for Authors
The article by Lewis et al. assesses the effectiveness of five automated MRI segmentation pipelines in comparison to manual methods for assessing GM1 gangliosidosis patients. The authors analyze which pipeline provides the most accurate results for evaluating various brain structures. The results are presented comprehensively, supplemented by graphs that clearly illustrate the differences between the various pipelines. They measured the volumes of different brain sections and found that the differences between the pipelines were minimal across all regions assessed. This consistency supports the reliability of these MRI analysis techniques, suggesting that radiologists can definitely use these automated methods to evaluate patient images in clinical practice. References were accurate and consistent. It is an interesting study.
Author Response
We appreciate Reviewer 2’s time and consideration of our study and acknowledgement of its importance.
Reviewer 3 Report
Comments and Suggestions for Authors
The authors present an interesting study in which various methods of automated analyses are compared to determine the accuracy and efficacy of each relative to one another in the context of data obtained from individuals with neurodegenerative diseases. Briefly, the authors utilise data from 24 individuals with neurodegenerative disorders and age matched controls in five systems to measure various areas of the brain. The strengths and weaknesses of each system were discerned, with strong data indicating that these previously questionable approaches could produce results comparable to that of manual analyses. These data suggest automated systems may be a more viable support for image analyses moving forward than was once thought, and warrant further development in the interest of improving patient diagnoses.
Overall I thought this was a well written, detailed study with only two minor points warranting attention. The authors should consider the following when submitting a resubmission.
- The design of Figure 1 should be revisited. It would for example be useful if the arrows from the boxes contained in the ‘middle column’ continued to the other side where ‘output volumetric data’ is positioned with the exception of the ‘manual’ and ‘FSL’ rows which have a third ‘step’. In its current form it looks incomplete and may cause confusion.
- For clarity, it would be advised that the n-number be included on all figures that contain data/statistics e.g. Figure 3 and 4.
Author Response
We would like to thank you for the acknowledgement of our work and appreciate the suggestions of how to improve our study.
Comment 1: The design of Figure 1 should be revisited. It would for example be useful if the arrows from the boxes contained in the ‘middle column’ continued to the other side where ‘output volumetric data’ is positioned with the exception of the ‘manual’ and ‘FSL’ rows which have a third ‘step’. In its current form it looks incomplete and may cause confusion.
Response 1: We appreciate this comment and have updated figure 1 to address this concern.
Comment 2: For clarity, it would be advised that the n-number be included on all figures that contain data/statistics e.g. Figure 3 and 4.
Response 2: We appreciate the comment and have added the n-number to the above referenced figures.
Reviewer 4 Report
Comments and Suggestions for Authors
I thank the opportunity to review the original manuscript entitled “A Case for Automated Segmentation of MRI data in milder neurodegenerative diseases”, sent for publication in NeuroSci. The authors performed an interesting manuscript in which they evaluated the potential use of five different automated segmentation pipelines with manual segmentation process using images from patients with GM1 gangliosidosis compared to neurotypical controls. The authors showed that similar profiles and results related to segmentation were observed with the automated techniques compared to manual process. Furthermore, they showed that for different brain regions and key structures (thalamus, lentiform nucleus, cerebellum) different quality of the observed profiles were identified, thus, disclosing the complexity related to the isolated use of these techniques without a neuroradiologist evaluation. This study brings important contributions also to a better understanding of the potential role of AI-based techniques for neuroimaging applications, such as in the natural history studies and correlating with clinical severity.
Some points need further discussion or clarification by the authors:
- Title: The title does not really give a proper idea of the manuscript content. For example, there is no mention to GM1 gangliosidosis, which is the prototype and focus during the analysis of this study. It should also be reviewed the aspect of describing in the title the disorder as “milder neurodegenerative diseases”. GM1 gangliosidosis represents an extremely complex inherited metabolic disorder with a severe neurological presentation.
- Have the authors evaluated any type of correlation related to symptoms presented by the patients (such as dystonia, parkinsonism, cerebellar ataxia, cognitive decline) and specific neuroimaging findings?
- Have the authors evaluated neuroimaging and clinical aspects considering the time since symptom-onset? In supplement C, age-matched methodology for juvenile patients is presented, and 6 individuals were from a group before age 11 years and 8 were from a group with 11 years or after – the reader does not know the time which passed since the onset of the first symptoms and signs of the disease.
- Do the authors think a similar process may be observed in other inherited neurometabolic disorders (such as GM2 gangliosidosis and ceroid neuronal lipofuscinosis)?
Author Response
We would like to thank you for the acknowledgement of our work and appreciate the suggestions of how to improve our study.
Comment 1: Title: The title does not really give a proper idea of the manuscript content. For example, there is no mention to GM1 gangliosidosis, which is the prototype and focus during the analysis of this study. It should also be reviewed the aspect of describing in the title the disorder as “milder neurodegenerative diseases”. GM1 gangliosidosis represents an extremely complex inherited metabolic disorder with a severe neurological presentation.
Response 1: We agree that GM1 as a whole is a severely progressive neurodegenerative disease. The word ‘milder’ was designated in relation to Type I GM1 gangliosidosis patients, but this study focuses solely on Type II patients. However, we acknowledge this is a relative comparison and have removed the word milder in the Title of the manuscript. The title now reads “A Case for Automated Segmentation of MRI Data in Neurodegenerative Diseases: Type II GM1 Gangliosidosis”.
Comment 2: Have the authors evaluated any type of correlation related to symptoms presented by the patients (such as dystonia, parkinsonism, cerebellar ataxia, cognitive decline) and specific neuroimaging findings?
Response 2: We have evaluated correlations in this cohort between MRI volumetric analysis and clinical symptoms in our previous study (doi:10.1016/j.ymgme.2025.109025). We have also added the following language to the introduction to establish the importance of these findings: “Furthermore, in our previous study, we showed reductions in the volumes of the total brain, caudate nucleus, thalamic nucleus, lentiform nucleus, corpus callosum, and ventricle enlargement correlated with patients’ clinical severity and ability to meet the demands of daily life, emphasizing volume-based morphology’s role as a sensitive imaging marker in this cohort.”
Comment 3: Have the authors evaluated neuroimaging and clinical aspects considering the time since symptom-onset? In supplement C, age-matched methodology for juvenile patients is presented, and 6 individuals were from a group before age 11 years and 8 were from a group with 11 years or after – the reader does not know the time which passed since the onset of the first symptoms and signs of the disease.
Response 3: We have previously reported on the correlation between clinical aspects and neuroimaging findings in this cohort (doi:10.1016/j.ymgme.2025.109025). However, we have now updated our manuscript to provide this detail in the introduction.
Furthermore, in our previous study, we showed volumetric reductions in the volumes of the total brain, caudate nucleus, thalamic nucleus, lentiform nucleus, corpus callosum, and ventricle enlargement correlated with patients’ clinical severity and ability to meet the demands of daily life, emphasizing volume-based morphology’s role as a sensitive imaging marker in this cohort.
We have also provided the age of diagnosis and age of symptom onset for the patients in Supplementary Table A1.
Comment 4: Do the authors think a similar process may be observed in other inherited neurometabolic disorders (such as GM2 gangliosidosis and ceroid neuronal lipofuscinosis)?
Response 4: We agree and have updated the discussion to include potential future use cases for this type of analysis with the following language: “For more common neurodegenerative diseases like Alzheimer’s and Parkinson’s, fully automated segmentation of MRI data utilizing Freesurfer, CAT12, or SPM12 are commonplace. However, this has not trickled down into rarer diseases where segmentation techniques are also applicable for evaluating disease progression and potential therapeutic benefit. For instance, automated segmentation would be amenable to juvenile and late-infantile GM2 gangliosidosis which has similar MRI progression to juvenile and late-infantile GM1 gangliosidosis. Furthermore, other lysosomal diseases like Gaucher, Fabry, and neuronal ceroid lipofuscinoses could be analyzed using this type of neuroimaging approach due to previously established brain involvement. This ultimately extends to cerebellar ataxias, Parkinsonism’s, and inherited prion diseases. With this in mind, researchers and clinicians should evaluate data of patients less than 2 years old with caution due to immature myelination. Studies aiming to evaluate this infant patient population should consider Infant Freesurfer or alternative pipelines specifically designed for analyzing infant MRI data.”
Round 2
Reviewer 1 Report
Comments and Suggestions for Authors
The revisions have been well incorporated. Thank you for your efforts.
Reviewer 4 Report
Comments and Suggestions for Authors
The authors properly discussed the points that were raised in the previous stage of review. The modifications made to the original text brought marked improvements in the discussion of the manuscript.